# Phubbed and Furious: Narcissists’ Responses to Perceived Partner Phubbing

**DOI:** 10.3390/bs15070853

**Published:** 2025-06-24

**Authors:** Claire M. Hart, Katherine B. Carnelley, Laura M. Vowels, Tessa Thejas Thomas

**Affiliations:** 1School of Psychology, Highfield Campus, University of Southampton, University Road, Southampton SO17 1BJ, UK; k.carnelley@soton.ac.uk (K.B.C.);; 2School of Psychology, Whitelands College, University of Roehampton, London SW15 4JD, UK; laura.vowels@roehampton.ac.uk

**Keywords:** grandiose narcissism, phubbing, daily diary, relationship satisfaction, retaliation

## Abstract

We conducted a diary study to examine how narcissism influences reactions to daily perceived partner phubbing (*N* = 196). We examined relationships between two facets of narcissism (rivalry and admiration) and personal and relational well-being, reactions to phubbing, reports of retaliation, and motives for retaliation. On average, participants higher in rivalry reported lower self-esteem and higher depressed and anxious mood, whilst participants higher in admiration reported greater relationship satisfaction, higher self-esteem, lower depressed and anxious mood, and lower levels of anger/frustration. These patterns held regardless of whether they were phubbed or not. In response to partner phubbing, participants higher in rivalry reported, on average, greater curiosity, resentment, conflict, and retaliation. On days when participants reported higher phubbing, those with higher rivalry reported greater curiosity, while those higher in admiration reported greater conflict. When retaliating to phubbing, those higher in rivalry did so, on average, to get back at their partner and to seek support and approval from others, whereas those higher in admiration were less likely to report boredom as a reason for retaliating. Our findings contribute to the sparse literature on narcissism and phubbing by showing how narcissists respond to being phubbed. We discuss how phubbing may exacerbate their relational difficulties.

## 1. Introduction

Perceived partner phubbing (a portmanteau of the words “phone” and “snubbing”) refers to the perception that a partner’s phone use decreases the face-to-face communication quality due to reduced partner attention ([50]). With technology becoming increasingly embedded in our daily interactions, it is important to understand its impact on the quality of in-person interactions, relationship dynamics, and well-being. Phubbing within romantic relationships has been shown to increase conflict, which in turn has been linked with lower relationship satisfaction and poorer well-being ([31]). Recent research has started to explore how couple members respond to perceived partner phubbing and the extent to which couple members engage in retaliatory phubbing themselves. [57] ([57]) found that when daily perceived partner phubbing was high, phubbees reported lower relationship satisfaction and greater feelings of anger, resentment, and retaliatory phone use, with the latter motivated by revenge and seeking support and approval from others. Our current study builds on this previous work by examining the individual difference variable of narcissism in responses to perceived partner phubbing. No research to date has examined narcissism and phubbing in romantic contexts. Using a daily diary approach, we explore whether narcissism (broken down into rivalrous and admirative facets) moderates the relationships between perceived phubbing and daily outcomes related to relational and personal well-being, phubbing responses, and retaliatory behaviours.

### 1.1. Emotional and Behavioural Responses to Perceived Partner Phubbing

Attention and responsiveness to a romantic partner are key predictors of relationship satisfaction (e.g., [4]; [39]). According to the displacement hypothesis, phubbing may interfere with the ability to attend to the partner and with the quality of time spent together ([2]). Furthermore, equity theory ([30]) posits that equal investment in a close relationship enhances relationship satisfaction. If an individual perceives unequal investment in their relationship, they may feel distressed and promote efforts to try and restore balance. Phubbing may signal such imbalance, as supported by research showing that individuals who perceive partner phubbing experience greater feelings of exclusion, diminished intimacy, and reduced partner responsiveness ([6]; [42]; [58]).

Perceived phubbing can also be conceptualised as a relational rupture—a moment in which shared presence and emotional reciprocity are broken. From the perspective of attachment theory, such interruptions may be experienced as a threat to relational security, particularly for individuals with anxious or avoidant attachment styles (e.g., [43]). The inattention implied by phubbing may signal to the phubbee that the partner is unavailable or disinterested, triggering emotional responses such as rejection sensitivity, protest behaviours, or withdrawal. In this sense, phubbing represents not only an imbalance in investment but also a failure of attunement, where one partner no longer prioritises mutual engagement—a core tenet of relational maintenance and intimacy.

The phubbee may perceive these negative evaluations as either due to shortcomings in the relationship, resulting in lower relationship satisfaction (e.g., [16]; [40]; [60]), or due to the phubbee’s personal inadequacies, resulting in distress. For example, the phubbee may feel that their partner does not consider them important or interesting enough to deserve attention ([16]), in turn lowering their self-esteem (e.g., [39]; [60]).

Perceptions of partner phubbing have been linked with lower relationship functioning in both cross-sectional and diary studies (e.g., [17]; [28]; [38]). Building on this work, [12] ([12]) explored how perceived phubbing and actual phubbing impacted relationship functioning in a dyadic diary and two months later. Their findings showed that daily perceived phubbing was associated with lower relationship quality, though its effects did not persist over the two-month period. Moreover, actual phubbing behaviour did not predict relationship quality either daily or two months later, underscoring the significance of perceptions in understanding these dynamics.

A recent scoping review on partner phubbing and mental health outcomes in romantic relationships ([33]) concluded that perceived partner phubbing adversely affects the phubbee’s personal well-being, contributing to declines in life satisfaction, increased depression, and heightened negative emotions, anxiety, and anger/frustration.

[60] ([60]) found that couple members who perceived greater partner phubbing also reported higher levels of depression, while [53] ([53]) found greater partner phubbing to be associated with increased anxiety. [45] ([45]) examined other emotional responses to perceived phubbing and found that the majority of participants reported feeling annoyed (83%) and angry (67%). In a diary study, [21] ([21]) observed significantly higher anger on days when participants reported perceived partner phubbing compared to days without it. [34] ([34]) also found that perceived phubbing correlated with heightened jealousy and anger, with jealousy negatively impacting relational cohesion. Despite these predominantly negative outcomes, some participants (38.1%) reported feeling indifferent to perceived phubbing, suggesting emotional responses to phubbing may vary across individuals.

Common responses to perceived phubbing include ignoring the behaviour ([31]), intervening, or retaliating by mimicking the partner’s phubbing behaviour ([34]). The motivations behind these responses have not been investigated extensively. Ignoring phubbing may be a passive strategy to avoid conflict, though doing so could impact the phubbee’s well-being negatively. Conversely, retaliation by mirroring the behaviour may be a tit-for-tat strategy driven by a desire for revenge. Research indicates that individuals engage in phubbing even while recognizing its annoyance to others ([1]). [57] ([57]) examined responses to perceived partner phubbing in a daily diary study and found that on days with higher perceived phubbing, phubbees reported increased curiosity about the phubbing behaviour, more resentment, and greater phubbing retaliation. They were among the first to examine motivations for retaliation, identifying revenge, need for support, and need for approval from others as key factors. The current study focuses on understanding these behavioural responses, particularly examining how narcissism may influence these reactions.

### 1.2. Narcissism and Its Role in Romantic Relationships

Grandiose narcissism is a personality trait characterised by an inflated, highly favorable self-image, a persistent need for attention and self-promotion, and a lack of concern for the needs or feelings of others ([11]). With their high feelings of entitlement and grandiosity ([54]), low commitment to relationships ([10]), and lack of empathy even for close others ([29]), grandiose narcissists are known for their troubled romantic relationships ([20]). They frequently use their relationships to fulfil their agentic desires, often resulting in short-lived ([61]), unsatisfying ([62]), and coercive ([9]) relationships. Given the prevalence of phone use in daily life, it is common for couple members to phub each other *and* be phubbed ([15]). However, research has yet to explore how narcissistic individuals respond to being phubbed within a relational context and how this may contribute to their relationship difficulties. This gap is important because narcissistic traits have been shown to influence reactions to self-esteem threats ([8]), perceived inequity in relationships ([52]), and experiences of jealousy in relational contexts ([14]), all of which are often implicated in phubbing incidents.

In this study, we used the Narcissistic Admiration and Rivalry Model (NARC; [5]) to explore the relationship among narcissism, perceived partner phubbing, and daily outcomes. The NARC model distinguishes between two facets of grandiose narcissism: narcissistic admiration and narcissistic rivalry. While both facets are associated with preserving grandiosity, they do so using differing strategies. Those high in narcissistic admiration employ prestige-based (agentic) strategies, such as self-promotion and charisma, to achieve status, whereas those high in narcissistic rivalry employ dominance-based (antagonistic) strategies, such as fear and aggression ([54]).

### 1.3. Narcissism and Phubbing: Research Gaps and Current Study Focus

The NARC model can be used to explain how trait narcissism interacts with perceived partner phubbing, depending on the individuals’ strategy for achieving status, which can shape the nature of the romantic relationship. For example, narcissistic admiration has been shown to be associated with better mate retention than narcissistic rivalry. Individuals high in narcissistic rivalry are more prone to cost-inflicting behaviours (e.g., threatening partner) as a strategy for mate retention ([64], [63]). Further, in a daily diary study, narcissistic rivalry was negatively associated with relationship satisfaction whereas narcissistic admiration was positively associated with relationship satisfaction ([49]). The findings were attributed to a difference in situation perception: individuals high in narcissistic admiration perceived situations as featuring more love and romance, whereas individuals high in narcissistic rivalry perceived more conflict and adversity in situations involving their significant other. Such findings suggest that individuals high in rivalry may engage with more maladaptive responses to perceived partner phubbing. Partner phubbing could be perceived as worse by individuals higher in narcissistic rivalry, compared to individuals lower in narcissistic rivalry, negatively influencing daily relationship satisfaction. Notably, both narcissistic admiration and rivalry predict conflict in relationships. Individuals high in admiration, although less inclined to do so, demonstrate cost-inflicting behaviours as a last resort ([63]). The effects are simply more evident in individuals high in rivalry ([27]).

Narcissism influences not only relational outcomes but also personal outcomes. In their cross-sectional data, [27] ([27]) found that narcissistic admiration was associated with higher positive affect and lower negative affect, whereas narcissistic rivalry had the opposite effect. Both admiration and rivalry have been positively associated with trait anger, although the association has been shown to be considerably higher for narcissistic rivalry ([5]). Moreover, admiration has been associated with greater emotion regulation while rivalry is associated with poorer emotion regulation ([13]). Consequently, individuals high in narcissistic rivalry may experience more intense negative emotions as an immediate response to relationship conflict. This may stem from differences in self-esteem; those high in rivalry often have fragile self-esteem, whereas individuals high in admiration display more stable and higher self-esteem ([5]). In support of this, [23] ([23]) observed stronger rivalry effects on reduced self-esteem following perceived social exclusion. These antagonistic traits may help explain why narcissistic rivalry is more detrimental to romantic relationships than admiration ([5]). In the context of phubbing as a form of exclusion, these findings suggest that rivalry and admiration may experience negative effects following from phubbing, but these may be experienced more acutely by those scoring higher in narcissistic rivalry.

To date, no studies have explored the relationship between narcissism and partner phubbing in a romantic context. Of the limited research that does exist on narcissism and general phubbing, researchers have primarily focused on whether narcissists are more likely to phub and why ([3]; [22]; [26]; [25]; [36]; [55]). Positive relationships between vulnerable narcissism and phubbing have been shown consistently, whereas the relationship between grandiose narcissism and phubbing has been mixed with research showing null findings, positive effects, and negative effects. Only two studies have examined narcissists as the recipient of general phubbing behaviours. [22] ([22]) found no relationship between narcissism and being phubbed. [55] ([55]) found no direct relationship between grandiose narcissism and reports of being phubbed, but did find an indirect relationship through heightened behavioural activation, such that those higher in grandiose narcissism were more likely to get phubbed due to their approach-oriented tendencies. No studies have examined how narcissists respond to being phubbed.

Therefore, we have yet to understand whether and how the phubbee’s narcissistic qualities affect perceptions and responses towards partner phubbing. In the present study, we are interested in whether individuals scoring higher in narcissistic admiration and rivalry react differently, both behaviourally and emotionally, to daily perceived partner phubbing. Additionally, we have yet to understand how this relationship then influences personal and relational outcomes, and we shall examine this within this study.

We posed the following research questions and tested pre-registered hypotheses concerning the role of narcissism in the present study. Note that Hypotheses 1, 2, and 3 are replications, while Hypotheses 4–7 are novel.

Research Question 1: How does narcissism influence one’s daily reports of relationship satisfaction, personal well-being, and anger/frustration?

Individuals scoring higher (versus lower) in narcissistic rivalry will report lower daily relationship satisfaction (Hypothesis 1a), whereas those scoring higher (versus lower) in narcissistic admiration will report higher relationship satisfaction (Hypothesis 1b).

Individuals scoring higher (versus lower) in narcissistic rivalry will report lower personal wellbeing (Hypothesis 2a), whereas those scoring higher (versus lower) in narcissistic admiration will report higher personal wellbeing (Hypothesis 2b).

Individuals scoring higher (versus lower) in narcissistic rivalry and admiration will report higher daily levels of anger/frustration (Hypothesis 3). These effects will be stronger for those scoring higher in narcissistic rivalry than admiration.

Research Question 2: Does narcissism moderate the relationship between daily perceived phubbing and relationship satisfaction, personal-well-being, anger, and response to being phubbed (curiosity, resentment, ignored, conflict, retaliation).[note 1]

On days when perceptions of phubbing are high, individuals higher in narcissistic rivalry and admiration will report lower relationship satisfaction (Hypothesis 4) and personal well-being (Hypothesis 5), higher anger/frustration (Hypothesis 6), and a greater likelihood of retaliatory behaviour (Hypothesis 7); however, the strength of these relationships between narcissism and outcomes will be greater for those scoring higher in rivalry than admiration.

Research Question 3: What motives underlie narcissists’ retaliation response to perceived partner phubbing?

Research question 3 was exploratory and aimed to identify participants’ motivations for engaging in retaliatory behaviours following phubbing. Given the limited prior research on this aspect, no directional hypotheses were proposed.

## 2. Method

### 2.1. Participants

We pre-registered this study on the Open Science Framework (https://osf.io/mdjn3) with a target sample size of *N* = 150, which we determined based on previous daily diary studies (e.g., [41]).

We advertised the study on social media (e.g., Facebook, Twitter, LinkedIn) and Prolific (https://www.prolific.com/). Our inclusion criteria required participants to (a) be aged 18 years or older, (b) currently be in a romantic relationship (with a minimum duration of six months), and (c) live with their partner. Participants were required to complete both a baseline survey and at least one diary survey. The initial sample consisted of 269 participants who completed the baseline survey. However, 73 participants did not continue the study beyond the baseline or their diary entries could not be matched to their baseline ID; these participants were thus excluded from analyses.

The final sample included 196 participants[note 2] (*M^age^* = 36.39, *SD* = 12.02; 112 participants recruited through Prolific and 82 participants recruited through social media).[note 3] Most identified as female (*n* = 144), with 50 identifying as male, 1 as non-binary, and 1 preferring not to say. The majority (*n* = 168) identified as straight, with smaller numbers identifying as bisexual (*n* = 8), lesbian (*n* = 14), gay (*n* = 3), or other (*n* = 3). Most participants were married (*n* = 100) or in a committed relationship (*n* = 90). Fewer participants selected the dating option (*n* = 5). Of the final sample, 49.5% had children. The average relationship length was 11.4 years (*SD* = 9.91).

Most participants were employed full-time (53.6%) or were students (11.2%). Others (9.2%) selected “other” for occupation, citing reasons such as working two jobs or living with a disability, while the remainder were part time employees (9.7%) or homemakers (3.6%).

### 2.2. Procedure

This study, advertised as a daily diary about “Mobile Phone Use in Romantic Relationships”, was granted ethical approval by the Faculty Ethics Committee (Psychology subcommittee) at the University of Southampton.

Participants began by reviewing the study information sheet. After providing informed consent, participants were asked to complete one baseline diary followed by nine short daily diaries, presented online via Qualtrics. Participants either stated their Prolific ID or an email address at each time point to allow us to align data across surveys. This information was also used to provide the participant with the follow up surveys (using either a custom allowlist or email list). Participants completed an average of 7.91 days, including baseline. The study took approximately one hour to complete across 10 days (5 min/day). Baseline measures (Day 1) included demographic questions (e.g., gender identity, sexual orientation, relationship status, relationship length, parental status, and occupation), narcissistic trait measures, and a measure of adult attachment2.[note 4] At the baseline assessment, participants who did not meet the inclusion criteria were thanked for their time, and the study was ended. The daily diaries measured perceived partner phubbing, responses to partner phubbing (curiosity, conflict, resentment, ignored, retaliation), relationship satisfaction, personal well-being (self-esteem, depressed and anxious mood), and anger/frustration. The order of presentation of daily measures was randomised to prevent order effects. Participants were requested to complete the daily diaries at the end of each day and to refrain from discussing responses with their partner. In the Prolific sample, daily surveys were only available from 4 pm to midnight before they expired.

Upon completion of baseline measures, participants recruited via social media were entered into a prize draw to win 1 of 3 GBP 50 Amazon vouchers and were told that each daily diary completed would result in an additional five raffle tickets. Participants recruited via Prolific were paid GBP 1.50 for completing the baseline survey and a further GBP 0.50 for each of the nine diary sessions. To be eligible for compensation, participants had to complete a minimum of 8 of the 10 surveys. On completing the study, participants were thanked and debriefed.

### 2.3. Measures

#### 2.3.1. Daily Perceived Phubbing

We assessed daily perceptions of partner phubbing (Pphubbing) with four of the original nine items of the Pphubbing Scale ([50]): “Today, my partner placed his/her mobile phone where they could see it when we were together”, “Today, my partner glanced at his/her mobile phone when talking to me”, “Today, when my partner’s phone rang or beeped, they pulled it out even if we were in the middle of a conversation”, and “Today, when my partner and I were together, my partner’s mobile phone use interfered with our interactions.” Participants reported their partner’s mobile phone use on a 5-point scale (1 = *not at all*, 2 = *a little*, 3 = *a moderate amount*, 4 = *a lot*, *5* = *a great deal*).

#### 2.3.2. Daily Relationship Satisfaction

We assessed daily romantic relationship satisfaction using the satisfaction subscale of the Perceived Relationship Quality Component Inventory (PRQC; [19]). Participants rated how satisfied, happy, and content they were with their relationship on that day on a 7-point scale, ranging from 1 (*not at all*) to 7 (*extremely*). A mean score of these three items was computed.

#### 2.3.3. Daily Self-Esteem

We assessed daily self-esteem with the Single-Item Self-Esteem Scale (SISE; [51]). Participants responded to the item, “Today, I have high self-esteem”, on a 7-point scale (1 = *not very true of me*, 7 = *very true of me*).

#### 2.3.4. Daily Depressed/Anxious Mood

We assessed daily reports of depressed and anxious mood with the 4-item Patient Health Questionnaire (PHQ-4; [35]). Original instructions were modified to focus on how participants felt that day, for example, “Today, have you been bothered by any of these problems?”. Items measured anxious mood (“Feeling nervous, anxious, or on edge”, “Not being able to stop or control worrying”) and depressed mood (“Feeling down, depressed, or hopeless”, “Little interest or pleasure in doing things”). Participants rated items on a 4-point scale (1 = *not at all*, 2 *= a little*, 3 *= a moderate amount*, 4 = *a great deal*). A mean score of depressed mood and anxious mood was computed.

#### 2.3.5. Daily Anger/Frustration

We created three items to assess state anger/frustration: “Today, I felt angry”, “Today, I felt irritated”, “Today, I felt annoyed”. Participants reported daily how they felt on a 5-point scale, ranging from 1 (*not at all*) to 5 (*a great deal)*. A mean score of these three items was computed.

#### 2.3.6. Daily Responses to Being Phubbed

A skip pattern was employed in the survey such that if participants reported any partner phubbing (indicated by a score greater than 1 on this item), we asked them six questions regarding how they responded to the phubbing: “I told them I was not happy” (conflict), “I asked them what they were looking at” (curiosity), “I argued with them about their phone use” (conflict), “I felt resentful about their phone use” (resentment), “I ignored their phone use” (ignored), and “I picked up my own phone and used it” (retaliation). Participants rated these items on a 9-point scale (1 = *not at all*, 5 = *a moderate amount*, 9 = *a great deal*). Because each item assessed a different construct, they were analysed separately, except for conflict, where we combined two conceptually similar items into a mean score to represent a conflict response to phubbing.

#### 2.3.7. Daily Motivations for Retaliation

A skip pattern was employed in the survey such that if a participant reported on the daily responses that they retaliated when they were phubbed (indicated by a score greater than 1 on this item, “I picked up my own phone and used it”), they were presented with an additional four items. Participants rated their reasons for picking up their own phone and using it on an 8-point scale (1 = *strongly disagree*, 8 = *strongly agree*). Items included the following: “To get back at my partner”, “I was bored”, “To seek support from others”, “To seek approval from others”. We examined each item separately in the statistical analysis.

#### 2.3.8. Narcissism

Participants completed the Narcissistic Admiration and Rivalry Questionnaire (NARQ; [5]). The scale comprises nine items assessing narcissistic admiration (e.g., “I deserve to be seen as a great personality”) and nine items assessing narcissistic rivalry (e.g., I enjoy it when another person is inferior to me”). Participants rated their agreement with items on an 8-point scale from 1 (*Strongly disagree*) to 8 (Strongly *agree*). We computed a mean score for each of the narcissistic dimensions.

#### 2.3.9. Data Analysis

The study employed a nested design with days (level 1) nested within individuals (level 2). We conducted hierarchical linear modelling (HLM) using the *lme4* package with R ([48]) to account for nesting. Level 1 variables represented within-person variations (i.e., daily perceived phubbing effects on daily relationship satisfaction, daily self-esteem, depressed and anxious mood, daily anger/frustration, and daily phubbing response behaviour), allowing us to examine daily fluctuations in emotions and behaviour based on perceived partner phubbing. Time (scaled to start at 0) was factored in as a covariate. Level 2 variables were narcissistic admiration and rivalry. We also examined cross-level interactions between daily perceived phubbing and narcissistic admiration and rivalry, respectively. We used Restricted Maximum Likelihood (REML) to address missing data. Due to the lack of consensus on effect size measures in multilevel modelling ([46]), we report marginal and conditional R-squared values ([44]). We estimate the reliability of time-varying variables using the *multilevel.reliablity* function from the *psych* package.

## 3. Results

During data cleaning, duplicate participant surveys for a single day were removed. Checks for normality revealed skewness for certain variables, including conflict, revenge, support, and approval (as motivation for retaliation). These variables were log-transformed prior to analysis. Daily phubbing was person-mean centred to test for within-person effects. Descriptive statistics for daily data across nine days are provided in Table 1. The reliability of average of all ratings across all items and times (fixed time effects) are also provided in Table 1, but other forms of reliability are available on the OSF project page in the results file for interested readers. Cronbach’s alphas are provided for narcissistic rivalry and admiration. For correlations between all variables, see Table 2. Multilevel models were conducted using the *lme4 R* package to analyse the nested data at the within-person level (level 1). All models included both random intercepts and random slopes. Data and R code, are available on the OSF project page: https://osf.io/ts6gw/?view_only=f4f774dbc6bc4de78eb7677c65dc4ab6.

### 3.1. Daily Relationship Satisfaction, Personal Well-Being, and Anger/Frustration

See Table 3 for the full results for relationship satisfaction, personal well-being, and anger/frustration variables. On days when participants reported higher levels of perceived partner phubbing, they also reported lower relationship satisfaction (*p* < 0.001), higher anxious mood (*p* = 0.002), higher depression (*p* = 0.018), and higher anger/frustration (*p* < 0.001).[note 5] Participants higher in narcissistic rivalry reported lower self-esteem (*p* = 0.002), higher depressed mood (*p* = 0.024), and higher anger (*p* < 0.001) on average. Participants higher in narcissistic admiration reported higher relationship satisfaction (*p* = 0.001) and self-esteem (*p* < 0.001) and lower anxiety (*p* = 0.032), depression (*p = 0*.001), and anger (*p* = 0.011) on average. There were no significant narcissism*perceived partner phubbing interactions. No other variables were significant.

### 3.2. Daily Responses to Being Phubbed

Responses to perceived partner phubbing were analysed, including curiosity about the partner’s phone use, feelings of resentment, ignoring partner phubbing, conflict related to partner phubbing, and retaliatory behaviours. See Table 4 for the full results for the daily responses to being phubbed. On days when participants reported higher levels of perceived partner phubbing, they also reported significantly greater curiosity (*p* < 0.001), resentment (*p* < 0.001), conflict (*p* < 0.001), and retaliation (*p* < 0.001). In contrast, they were significantly less likely to feel ignored (*p* = 0.002).[note 6] Individuals higher in narcissistic rivalry also reported significantly higher curiosity (*p* = 0.008), resentment (*p* < 0.001), conflict (*p* = 0.033), and retaliation (*p* = 0.030). There were two significant interactions between narcissistic rivalry and perceived partner phubbing in predicting curiosity (*p* = 0.026) and between narcissistic admiration and perceived partner phubbing in predicting conflict (*p* = 0.008). Simple slopes analyses showed that participants higher in narcissistic rivalry (1 SD above the mean) reported significantly higher curiosity (*B* = 1.41, *t* = 11.92, *p* < 0.001) on days when they perceived their partner as phubbing them more (see Figure 1). Participants lower in narcissistic rivalry (1 SD below the mean) also reported higher curiosity on days when they perceived their partner as phubbing them more (*B* = 1.01, *t* = 8.35, *p* < 0.001). The difference in curiosity levels between participants higher and lower in narcissistic rivalry was only significant on days when perceived phubbing was higher, such that those higher in narcissistic rivalry reported greater curiosity than those lower in narcissistic rivalry (*B* = 0.445, *t* = 3.50, *p* < 0.001). No other variables were significant.

Simple slopes analyses showed that participants higher in narcissistic admiration (1 SD above the mean) reported significantly higher conflict (*B* = 0.292, *t* = 11.01, *p* < 0.001) on days when they perceived their partner as phubbing them more (see Figure 2). Participants lower in narcissistic admiration (1 SD below the mean) also reported higher conflict on days when they perceived their partner as phubbing them more (*B* = 0.188, *t* = 7.64, *p* < 0.001). The difference in conflict levels between participants higher and lower in narcissistic admiration was only significant on days when perceived phubbing was higher, such that those higher in narcissistic admiration reported greater conflict than those lower in narcissistic admiration (*B* = 0.474, *t* = 1.98, *p* = 0.049). No other variables were significant.

### 3.3. Daily Motivations for Retaliation

To gain insight into why phubbees may engage in retaliatory behaviour (i.e., picking up one’s own phone and using it as a response to perceived partner phubbing), we explored four potential motives: revenge, boredom, need for support, and need for approval. See Table 5 for the full results for daily motivations for retaliation. On days when participants reported higher levels of perceived partner phubbing, they also reported significantly higher agreement with all of the motivations: revenge (*p* < 0.001), boredom (*p* < 0.001), need for support (*p* = 0.004), and need for approval (*p* < 0.001).[note 7] Individuals higher in narcissistic rivalry were more likely to report getting back at their partner (*p* = 0.005), need for support (*p* = 0.004), and approval (*p* < 0.001) as reasons for retaliating. Individuals higher in narcissistic admiration were significantly less likely to report boredom as a motivator for retaliation (*p* = 0.031). No other variables were significant, and there were no significant narcissism*perceived partner phubbing interactions.

## 4. Discussion

### 4.1. Daily Relationship Satisfaction, Personal Well-Being, and Anger/Frustration

We investigated how trait narcissism was associated with daily reports of relationship satisfaction, personal well-being (self-esteem, anxious and depressed mood), and anger/frustration. Consistent with previous research ([32]; [49]; [61]), we found support for Hypothesis 1b; that narcissistic admiration was associated with higher relationships satisfaction, but no support for Hypothesis 1a; that narcissistic rivalry was associated with lower daily relationship satisfaction. Although results for Hypothesis 1a were in the expected direction, they failed to reach statistical significance. With more power, these results may emerge in line with previous diary studies ([32]; [49]; [61]).

Consistent with research linking narcissistic rivalry to poorer wellbeing outcomes and narcissistic admiration to better wellbeing outcomes ([18]; [23]; [27]; [37]), we found that narcissistic rivalry was associated with lower self-esteem and higher anxious and depressed mood, while narcissistic admiration was associated with higher self-esteem and lower anxious and depressed mood. These results support Hypotheses 2a and 2b.

Regarding aggression/frustration, we hypothesized that narcissistic rivalry and admiration would be associated with higher daily reports of anger/frustration (Hypothesis 3). Our results provided partial support for this hypothesis: narcissistic rivalry was positively associated with daily anger/frustration, while narcissistic admiration was negatively associated with these emotions. Although we anticipated that individuals high in admiration might experience negative emotions, especially in response to phubbing that threatens their need for attention and affirmation, research has shown that narcissistic admiration is linked to better emotion regulation, whereas narcissistic rivalry is associated with emotion dysregulation. [13] ([13]) showed that individuals with higher narcissistic admiration scores reported more effective responses to emotional challenges and greater emotional clarity than those scoring higher in narcissistic rivalry. In contrast, those with higher narcissistic rivalry scores reported greater difficulty in regulating and controlling emotional impulses. The pursuit of grandiosity in narcissistic admiration is largely associated with self-enhancement ([5]). As such, admirative narcissists may use charm-based tactics over dominance-based tactics, such as aggression, which may help explain the negative association with anger/frustration in our findings.

We explored narcissism as a potential moderator of daily perceptions of partner phubbing and daily personal/relational outcomes. Contrary to expectations, neither narcissistic admiration nor rivalry moderated the effects of perceived partner phubbing on relationship satisfaction, personal well-being (i.e., self-esteem, anxious mood, depressed mood), or feelings of anger and frustration. Thus, while narcissism independently showed associations with personal and relational outcomes, it did not significantly change how partner phubbing was related to these outcomes on a daily basis, offering no support for Hypotheses 4, 5, or 6. Thus, although increased perceptions of partner phubbing are associated with poorer relational and personal wellbeing, and narcissistic rivalry in particular is associated with poorer relational and personal wellbeing outcomes, narcissism does not exacerbate these relationships.

### 4.2. Daily Responses to Being Phubbed

In response to partner phubbing, we found that participants higher in narcissistic rivalry reported, on average, greater curiosity, resentment, conflict, and retaliation. Narcissistic admiration was not significantly associated with any of the daily responses. We also explored whether narcissistic traits moderated daily responses to phubbing. Contrary to Hypothesis 7, neither admiration nor rivalry moderated the relationship between perceived partner phubbing and retaliatory behaviours. However, significant interactions did emerge for other responses to being phubbed.

Research suggests that narcissism, along with other dark triad traits (psychopathy, Machiavellianism), correlate with increased engagement in romantic revenge ([7]). Specifically, narcissistic rivalry is characterized by maladaptive responses to perceived relational conflicts ([49]; [63]), and individuals high in narcissistic rivalry are known to engage in more romantic revenge ([5]). Our findings extend previous research on narcissism and romantic retaliation (e.g., [7]; [63]), suggesting that narcissistic rivalry is linked to maladaptive relational conflict strategies. In particular, curiosity about a partner’s phone use was more frequent on days with higher perceived phubbing among individuals high in narcissistic rivalry. We interpret this not as benign interpersonal interest but as a potential mate-guarding response—an effort to monitor and control the partner’s behaviour in response to perceived relational threat. This aligns with research suggesting that narcissistic rivalry involves heightened jealousy, vigilance, and threat sensitivity ([5]; [63]). Such behaviours may function to preserve the individual’s social status or relational dominance when their sense of superiority is challenged. Increased surveillance, including questioning a partner’s phone use, may reflect a need to reaffirm the partner’s loyalty or redirect their attention. Thus, curiosity may serve a defensive, strategic function in the face of perceived neglect.

We also found that participants higher in narcissistic admiration reported greater conflict on days with higher perceived partner phubbing. Although generally less reactive to conflict ([63]), individuals high in admiration may engage in disagreement when their need for attention and validation is frustrated. The absence of a main effect suggests that emotion regulation may typically buffer their responses; however, under threat, this regulation may fail. While both admiration and rivalry are associated with interpersonal conflict ([27]), our results suggest that rivalry is linked to conflict regardless of phubbing perceptions, whereas admiration appears to amplify conflict only under perceived threat. For those high in admiration, conflict may act as a means to regain control and reaffirm their value when they feel devalued.

### 4.3. Daily Motivations for Retaliation

For individuals who reported using their own phone as a form of retaliation against partner phubbing, we explored their motivations for engaging in this behaviour. Such reasons included getting back at their partner (revenge), boredom, or seeking support or approval from others. We found that individuals scoring higher in narcissistic rivalry were more likely to retaliate to seek revenge and to seek support and approval from others, whereas those higher in narcissistic admiration were less likely to report boredom as a motive.

Retaliation among those high in rivalry may reflect a defensive strategy to restore self-esteem or assert relational control following perceived rejection. Consistent with the literature ([5]), rivalry is characterised by hypersensitivity to ego threat, antagonism, and a desire to avoid subordination. Retaliatory phubbing may therefore function as a self-protective response, reasserting dominance or expressing frustration while mitigating emotional vulnerability.

This behaviour may also serve as a form of boundary-setting, particularly when attention needs are perceived as unmet. Phubbing one’s partner in return may communicate dissatisfaction nonverbally—avoiding direct confrontation while signalling hurt or disapproval. In this sense, retaliatory phubbing may operate as a passive form of protest.

Further, retaliation may reflect indirect communication strategies. For individuals who struggle with direct expression of emotional needs, especially those high in rivalry, retaliatory phubbing may offer a less risky route to signal rejection, anger, or frustration. Our finding that those higher in rivalry were more likely to seek external approval also suggests that retaliation may serve to reaffirm self-worth through alternative social channels.

Although narcissistic admiration was not linked to revenge-based retaliation, individuals high in admiration may avoid retaliating out of boredom to preserve their self-image or project emotional composure. This distinction further underscores the different self-regulatory aims of the two narcissism dimensions: while rivalry is driven by self-protection, admiration is guided by self-enhancement. Although we did not observe significant moderation effects of narcissism on specific retaliation motives, future work could examine whether retaliatory phubbing serves communicative, regulatory, or status-restoring functions depending on personality traits. Such research would clarify whether these behaviours are adaptive or maladaptive and illuminate the psychological mechanisms that underlie them.

### 4.4. Strengths, Limitations, and Future Directions

This study had several strengths. As a diary study, it captured daily responses to perceived phubbing, reducing the reliance on retrospective accounts and minimising the risk of biased responding. The sample was also sufficiently powered to test the hypotheses, which were pre-registered. We examined novel, theoretically driven hypotheses about the role of narcissism in response to phubbing, thereby enhancing our understanding of this potential threat to relationship dynamics. Further, by considering the admirative and rivalrous facets of narcissism concurrently, we were able to provide a more nuanced understanding of the influence of narcissism on perceived partner phubbing and outcomes.

Although this study focused on grandiose narcissism, future research should explore how vulnerable narcissism—characterised by hypersensitivity, insecurity, and a heightened need for reassurance ([47])—may shape responses to phubbing. Individuals high in vulnerable narcissism may interpret partner phubbing as confirming their fears of unworthiness or relational instability, leading to distress and avoidance rather than confrontation. Instead of retaliating or directly challenging their partner, they may withdraw emotionally, internalise feelings of rejection, or increase efforts to seek reassurance and validation. Passive-aggressive behaviours such as sulking, indirect communication, or overcompensation through people-pleasing may also emerge. In some cases, phubbing may prompt heightened emotional reactivity or rumination, reinforcing self-doubt and relationship insecurity. These more internalising, appeasing, or indirect responses contrast with the more externalising and dominance-driven responses typically associated with grandiose narcissism. In the current study, we examined a range of common responses drawn from the previous literature on interpersonal conflict and phubbing—including curiosity, conflict, resentment, ignoring the behaviour, and retaliation. We acknowledge that these behaviours do not capture the full spectrum of possible reactions to partner phubbing. Exploring a broader set of responses—including more covert, emotional, or long-term behavioural patterns—would be a valuable direction for future research, especially in relation to traits such as vulnerable narcissism.

Previous research ([15], [16]), as well as the present study, has demonstrated that an individual can simultaneously be both the phubber and the phubbee. While this study primarily focused on responses to being phubbed to build on existing research regarding narcissism as a predictor of phubbing behaviours ([3]; [22]; [26]; [25]; [36]; [55]), future research should consider examining both roles concurrently. Longitudinal studies involving both partners would provide a more holistic understanding of how narcissistic individuals engage in and respond to phubbing. Such an approach could offer valuable insights into the bidirectional dynamics of phubbing in relationships and facilitate the development of tailored interventions aimed at improving communication and relationship satisfaction.

In addition, the study had some limitations. It relied on self-reports, which may be influenced by memory and social desirability biases. Future research could benefit from exploring whether psychophysiological indicators of emotion, such as heart rate and skin conductance, are affected by perceived phubbing and how these measures align with self-report measures. Another limitation was that we did not test causal processes.: Future studies should consider observing couple interactions and behavioural responses to phubbing, as well as manipulating partner phubbing to test causality.

Although our study involved a substantial sample size, it was predominantly composed of women and individuals identifying as heterosexual. This lack of gender and sexual orientation diversity presents a notable limitation, as gendered norms and expectations can influence both the perception of phubbing and the expression of narcissistic traits. For instance, research indicates that perceived partner responsiveness was a significant mediator between perceived partner phubbing and relationship quality for women but not for men ([59]). Additionally, studies have shown that men typically score higher on grandiose narcissism, while women tend to score higher on vulnerable narcissism ([24]). Consequently, their behavioural and emotional responses to partner phubbing may differ, with potential implications for the interpretation of the results presented here (predominantly women). Moreover, relationship dynamics in predominantly heterosexual samples may not fully capture the experiences within LGBTQ+ relationships, where different social and relational contexts could influence perceptions of phubbing ([56]). For example, factors such as minority stress, communication patterns, and relational expectations may shape how phubbing is understood and its impact on relationship satisfaction and conflict resolution. Future research should therefore aim to replicate these findings in more diverse samples and explicitly explore how gender and sexual orientation influence the experience and impact of phubbing within romantic relationships, thus providing a more nuanced understanding of these dynamics.

## 5. Conclusions

This study highlights the role of narcissistic traits in shaping responses to partner phubbing, revealing nuanced differences between the rivalry and admiration facets of grandiose narcissism. By employing a diary study design, we were able to capture daily variations in personal and relational well-being and reactions to perceived partner phubbing. Our findings indicate that individuals higher in narcissistic rivalry experience lower self-esteem and heightened negative emotions, such as depression and anxiety, and, in response to being phubbed display increased curiosity, resentment, conflict, and retaliation, with the latter being driven by revenge and a desire for support and approval. In contrast, those higher in narcissistic admiration tend to experience greater relationship satisfaction, higher self-esteem, and lower depression and anxiety and anger/frustration. These individuals are less likely to retaliate to perceived partner phubbing due to boredom. These patterns remained consistent regardless of whether they were being phubbed or not. Notably, on days when participants reported higher phubbing, those with higher narcissistic rivalry reported greater curiosity, while those higher in narcissistic admiration reported greater conflict.

Exploring the role of narcissism in response to being phubbed helps to elucidate the complex interplay between personality traits and relational behaviours, providing insights into ways of fostering healthier relationships in the context of modern digital interactions. Understanding these dynamics is essential for developing targeted interventions aimed at mitigating the relational difficulties that narcissistic individuals may face in an era increasingly influenced by technology-mediated communication.

## Figures and Tables

**Figure 1 behavsci-15-00853-f001:**
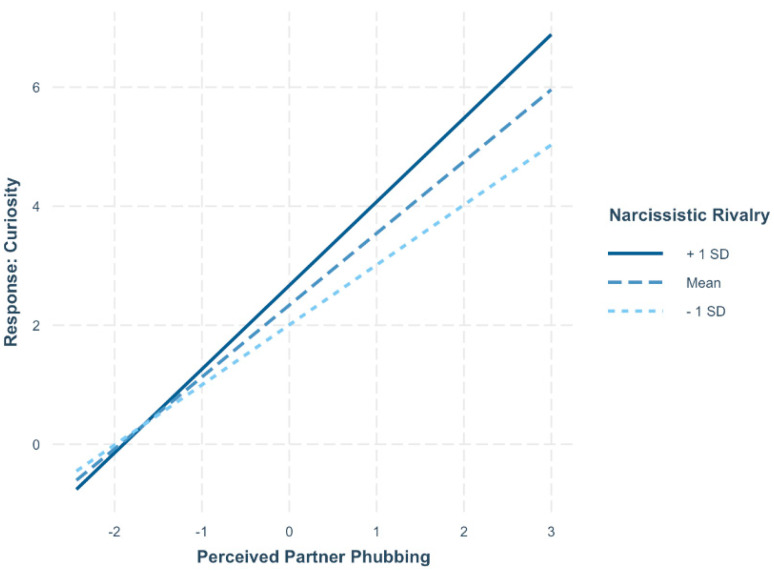
Interaction between narcissistic rivalry and perceived partner phubbing in predicting curiosity. On days when individuals perceived more phubbing by their partner, both those high and low in narcissistic rivalry reported increased curiosity. However, the effect was stronger for individuals higher in narcissistic rivalry, who reported significantly more curiosity than their lower-rivalry counterparts under high phubbing conditions.

**Figure 2 behavsci-15-00853-f002:**
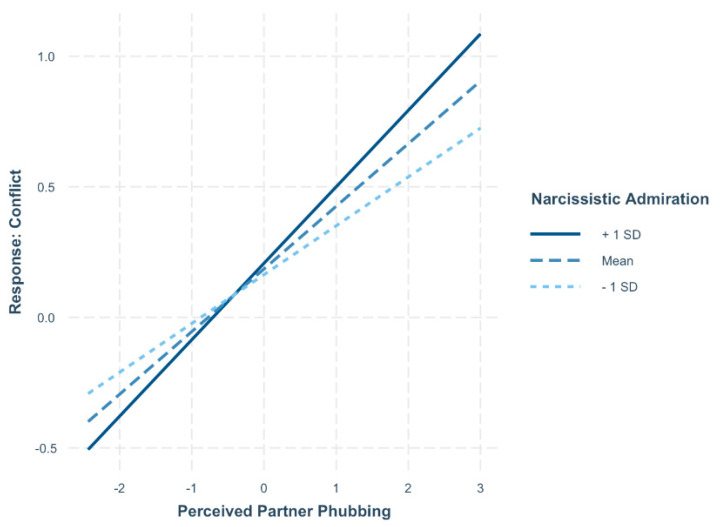
Interaction between narcissistic admiration and perceived partner phubbing in predicting conflict. On days when individuals perceived more phubbing by their partner, both those high and low in narcissistic admiration reported increased conflict. However, this association was stronger for individuals higher in admiration, who experienced significantly greater conflict than their lower-admiration counterparts under high phubbing conditions.

**Table 1 behavsci-15-00853-t001:** Descriptive statistics for all baseline and daily measures.

Variable	M	SD	Skewness	Kurtosis	Reliability
*Level 1*					
Perceived partner phubbing	2.45	0.80	0.43	0.03	0.96
Response: Curiosity	3.12	2.40	0.87	−0.52	
Response: Resentment	2.14	2.02	1.87	2.51	
Response: Ignored	5.84	2.71	−0.50	−1.07	
Response: Conflict *	1.40	1.22	2.50	6.37	
Response: Retaliation	4.64	2.65	−0.06	−1.34	
Motivation for retaliation: Revenge	1.95	1.67	1.71	1.77	
Motivation for retaliation: Boredom	5.43	1.97	−0.69	−0.24	
Motivation for retaliation: Support *	1.50	1.30	3.32	11.24	
Motivation for retaliation: Approval *	1.38	1.02	3.67	15.32	
Relationship satisfaction	5.70	1.25	−1.10	1.10	1.00
Anger/frustration	1.72	0.84	1.51	2.20	0.98
Self-esteem	4.12	1.74	−0.18	−0.89	
Anxious mood	1.68	0.80	1.19	0.68	0.98
Depressed mood	1.64	0.78	1.23	0.87	0.98
*Level 2*					
Narcissistic rivalry	2.29	1.03	1.19	1.47	0.85
Narcissistic admiration	3.64	1.23	0.40	−0.06	0.86

* Log transformed for analyses.

**Table 2 behavsci-15-00853-t002:** Correlations between variables.

Variable	1	2	3	4	5	6	7	8	9	10	11	12	13	14	15	16
1. Rivalry																
2. Admiration	0.35 **															
3. Phubbing	0.04	0.03														
4. Relationship satisfaction	−0.01	0.12	−0.24 **													
5. Self-esteem	0.01	0.50 **	−0.01	0.12												
6. Anxious mood	0.08	−0.10	0.14	−0.24 **	−0.38 **											
7. Depressive mood	0.14 *	−0.18 *	0.05	−0.25 **	−0.48 **	0.68 **										
8. Anger/frustration	0.12	−0.03	0.25 **	−0.40 **	−0.14	0.55 **	0.49 **									
9. Response: Curiosity	0.24 **	0.00	0.27 **	−0.04	−0.15 *	0.15 *	0.12	0.15 *								
10. Response: Resentment	0.17 *	−0.07	0.49 **	−0.38 **	−0.18 *	0.36 **	0.33 **	0.32 **	0.39 **							
11. Response: Ignored	−0.09	−0.01	−0.06	0.05	0.01	−0.01	0.03	−0.08	−0.11	−0.28 **						
12. Response: Conflict	0.17 *	0.09	0.29 **	−0.17 *	−0.01	0.16 *	0.18 *	0.31 **	0.50 **	0.58 **	−0.18 *					
13. Response: Retaliation	0.15 *	−0.04	0.29 **	−0.18 *	−0.20 **	0.15 *	0.20 **	0.14	0.32 **	0.19 **	0.22 **	0.18 *				
14. Revenge	0.21 **	0.06	0.12	−0.37 **	0.00	0.18 *	0.17 *	0.28 **	0.32 **	0.50 **	−0.18 *	0.49 **	0.15			
15. Boredom	0.15	−0.12	0.22 **	−0.19 *	−0.27 **	0.16	0.32 **	0.26 **	0.27 **	0.22 **	0.04	0.25 **	0.50 **	0.33 **		
16. Support	0.23 **	0.08	0.19 *	−0.24 **	0.00	0.29 **	0.25 **	0.24 **	0.22 **	0.42 **	0.03	0.35 **	0.24 **	0.44 **	0.23 **	
17. Approval	0.30 **	0.14	0.12	−0.21*	0.04	0.24 **	0.23 **	0.29 **	0.20 *	0.39 **	−0.09	0.45 **	0.20 *	0.53 **	0.17 *	0.76 **

*Note*. * indicates *p* < 0.05. ** indicates *p* < 0.01.

**Table 3 behavsci-15-00853-t003:** Perceived partner phubbing, narcissistic rivalry, and narcissistic admiration predicting relationship satisfaction, personal well-being, and anger/frustration.

Relationship Satisfaction				Self-Esteem			Anxious Mood			Depressed Mood			Anger/Frustration		
*Predictors*	*Estimates*	*CI*	*p*	*Estimates*	*CI*	*p*	*Estimates*	*CI*	*p*	*Estimates*	*CI*	*p*	*Estimates*	*CI*	*p*
Intercept	5.82	5.65–5.98	**<0.001**	4.07	3.87–4.26	**<0.001**	1.64	1.55–1.73	**<0.001**	1.56	1.47–1.66	**<0.001**	1.65	1.56–1.74	**<0.001**
Partner Phubbing	−0.17	−0.23–−0.11	**<0.001**	−0.06	−0.13–0.01	0.109	0.07	0.03–0.11	**0.002**	0.05	−0.01–0.09	**0.018**	0.16	0.11–0.22	**<0.001**
Narcissistic Rivalry	−0.15	−0.31–0.01	0.075	−0.31	−0.38– −0.11	**0.002**	0.08	−0.00–0.16	0.051	0.09	0.01–0.17	**0.024**	0.13	0.05–0.21	**0.001**
Narcissistic Admiration	0.23	0.09–0.37	**0.001**	0.82	0.66–0.99	**<0.001**	−0.08	−0.15–−0.01	**0.032**	−0.11	−0.18–−0.04	**0.001**	−0.09	−0.15–−0.02	**0.011**
Time	−0.01	−0.03–0.01	0.349	0.01	−0.01–0.03	0.199	−0.02	−0.03–−0.01	**0.001**	−0.02	−0.03–−0.00	**0.009**	−0.01	−0.03–−0.00	0.053
Pphubbing*Narcissistic Rivalry	−0.03	−0.09–0.04	0.459	−0.05	−0.13–0.02	0.165	0.01	−0.04–0.06	0.725	0.02	−0.03– 0.07	0.388	0.03	−0.05–0.07	0.733
Pphubbing*Narcissistic Admiration	−0.00	−0.06–0.05	0.882	−0.02	−0.08–0.05	0.615	0.02	−0.02–0.06	0.323	0.01	−0.03–0.05	0.676	0.05	−0.00–0.10	0.051
Random Effects															
σ^2^	0.43			0.59			0.25			0.21			0.37		
τ_00_	1.23_ParticipantID_			1.69_ParticipantID_			0.33_ParticipantID_			0.36_ParticipantID_			0.25_ParticipantID_		
τ_11_	0.01_ParticipantID.time_			0.01_ParticipantID.time_			0.00_ParticipantID.time_			0.00_ParticipantID.time_			0.00_ParticipantID.time_		
ρ_01_	−0.04_ParticipantID_			0.11_ParticipantID_			−0.32_ParticipantID_			−0.44_ParticipantID_			−0.18_ParticipantID_		
ICC	0.76			0.77			0.55			0.60			0.41		
N	210_ParticipantID_			209_ParticipantID_			210_ParticipantID_			210_ParticipantID_			210_ParticipantID_		
Observations	1514			1514			1517			1502			1512		
Marginal R^2^/Conditional R^2^	0.044/0.772			0.262/0.828			0.027/0.566			0.040/0.612			0.048/0.442		

*Note.* Bolded values are statistically significant and highlighted to aid interpretation.

**Table 4 behavsci-15-00853-t004:** Perceived partner phubbing, narcissistic rivalry, and narcissistic admiration predicting daily responses to being phubbed.

Curiosity				Resentment			Ignored			Conflict			Retaliation		
*Predictors*	*Estimates*	*CI*	*p*	*Estimates*	*CI*	*p*	*Estimates*	*CI*	*p*	*Estimates*	*CI*	*p*	*Estimates*	*CI*	*p*
Intercept	2.45	2.18–2.71	**<0.001**	1.69	1.47–1.91	**<0.001**	5.83	5.50–6.16	**<0.001**	0.17	0.11–0.23	**<0.001**	3.82	3.51–4.13	**<0.001**
Partner Phubbing	1.21	1.06–1.36	**<0.001**	1.00	0.88–1.12	**<0.001**	−0.39	−0.64–−0.15	**0.002**	0.24	0.21–0.27	**<0.001**	1.01	0.84–1.18	**<0.001**
Narcissistic Rivalry	0.31	0.08–0.54	**0.008**	0.34	0.15–0.52	**<0.001**	−0.23	−0.52–0.07	0.132	0.06	0.00–0.11	**0.033**	0.32	0.03–0.61	**0.030**
Narcissistic Admiration	−0.05	−0.25–0.15	0.635	−0.08	−0.24–0.08	0.342	−0.06	−0.31–0.20	0.664	0.02	−0.03–0.06	0.437	−0.05	−0.30–0.20	0.686
Time	−0.03	−0.06–0.01	0.115	0.00	−0.03–0.03	0.976	−0.08	−0.14–−0.02	**0.011**	0.00	−0.00–0.01	0.254	−0.13	−0.18–−0.09	**<0.001**
Pphubbing*Narcissistic Rivalry	0.19	0.02–0.35	**0.026**	0.09	−0.04–0.22	0.158	−0.13	−0.41–0.14	0.347	0.02	−0.01–0.06	0.171	−0.03	−0.22–0.17	0.780
Pphubbing*Narcissistic Admiration	0.04	−0.11–0.18	0.633	−0.09	−0.21–0.03	0.130	0.13	−0.11–0.37	0.295	0.04	0.01–0.07	**0.008**	0.04	−0.13–0.21	0.638
Random Effects															
σ^2^	2.04			1.15			5.60			0.09			2.60		
τ_00_	2.58_ParticipantID_			1.97_ParticipantID_			2.63_ParticipantID_			0.14_ParticipantID_			3.67_ParticipantID_		
τ_11_	0.01_ParticipantID.time0_			0.02_ParticipantID.time0_			0.04_ParticipantID.time0_			0.00_ParticipantID.time0_			0.03_ParticipantID.time0_		
ρ_01_	−0.43_ParticipantID_			−0.50_ParticipantID_			0.07_ParticipantID_			−0.46_ParticipantID_			−0.16_ParticipantID_		
ICC	0.53			0.59			0.39			0.58			0.60		
N	210_ParticipantID_			209_ParticipantID_			210_ParticipantID_			210_ParticipantID_			210_ParticipantID_		
Observations	1226			1223			1225			1224			1226		
Marginal R^2^/Conditional R^2^	0.126/0.588			0.138/0.644			0.018/0.404			0.109/0.628			0.088/0.635		

*Note.* Bolded values are statistically significant and highlighted to aid interpretation.

**Table 5 behavsci-15-00853-t005:** Perceived partner phubbing, narcissistic rivalry, and narcissistic admiration predicting daily motivations for retaliation.

Revenge				Boredom			Need for Support			Need for Approval		
*Predictors*	*Estimates*	*CI*	*p*	*Estimates*	*CI*	*p*	*Estimates*	*CI*	*p*	*Estimates*	*CI*	*p*
Intercept	0.29	0.22–0.36	**<0.001**	4.97	4.67–5.27	**<0.001**	0.21	0.14–0.28	**<0.001**	0.16	0.10–0.21	**<0.001**
Partner Phubbing	0.19	0.14–0.25	**<0.001**	0.53	0.34–0.73	**<0.001**	0.07	0.02–0.11	**0.004**	0.09	0.05–0.13	**<0.001**
Narcissistic Rivalry	0.10	0.03–0.17	**0.005**	0.20	−0.07–0.48	0.143	0.09	0.03–0.15	**0.004**	0.13	0.08–0.18	**<0.001**
Narcissistic Admiration	0.02	−0.04–0.08	0.478	−0.27	−0.51–−0.02	**0.031**	0.03	−0.02–0.08	0.233	0.02	−0.03–0.06	0.447
Time	−0.02	−0.03–−0.01	**0.001**	−0.09	−0.14–−0.04	**0.001**	−0.01	−0.02–0.00	0.249	−0.00	−0.02–0.01	0.371
Pphubbing*Narcissistic Rivalry	0.03	−0.03–0.09	0.276	0.08	−0.15–0.30	0.507	−0.02	−0.07–0.03	0.428	−0.02	−0.07–0.02	0.284
Pphubbing*Narcissistic Admiration	−0.02	−0.08–0.04	0.466	0.13	−0.09–0.35	0.239	−0.00	−0.06–0.05	0.809	−0.00	−0.05–0.04	0.903
Random Effects												
σ^2^	0.14			1.86			0.11			0.08		
τ_00_	0.15_ParticipantID_			2.57_ParticipantID_			0.11_ParticipantID_			0.10_ParticipantID_		
τ_11_				0.03_ParticipantID.time0_			0.00_ParticipantID.time0_			0.00_ParticipantID.time0_		
ρ_01_				−0.21_ParticipantID_			−0.26_ParticipantID_			−0.35_ParticipantID_		
ICC	0.52			0.59			0.50			0.55		
N	185_ParticipantID_			185_ParticipantID_			184_ParticipantID_			184_ParticipantID_		
Observations	736			737			734			734		
Marginal R^2^/Conditional R^2^	0.098/0.566			0.054/0.612			0.59/0.527			0.126/0.607		

*Note.* Bolded values are statistically significant and highlighted to aid interpretation.

## Data Availability

Data and code for this study can be found here: https://osf.io/ts6gw/?view_only=f4f774dbc6bc4de78eb7677c65dc4ab6.

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
