# Peer review of "Phubbed and Furious: Narcissists’ Responses to Perceived Partner Phubbing"

_behavsci, 2025, doi:10.3390/bs15070853_

Round 1

Reviewer 1 Report

Comments and Suggestions for Authors

I would like to thank the authors for the opportunity to read their manuscript on narcissism and phubbing. My comments are below:

  • I like the intro (not all comments have to be negative 😊)
  • If the RQ3 is exploratory, it might help readers to state that below the question.
  • “Fewer students selected the dating option” <- students or participants?
  • There’s a typo in “Pphubbing scale” in the methods section
  • When reporting measures, the authors should also reflect on their metric properties (at least internal consistency if not model fit). This is mentioned later in the Results, but the methods section is where you have to justify the choice of measures, so it might fit better there.
  • What was the correlation between narcissistic rivalry and admiration?
  • An insignificant p has been bolded in Table 4
  • Causal language is a bit stronger in the Discussion than I’d prefer, considering potential Granger causality (e.g., rows 459-460). Some readers may be unfamiliar with research methodology and consider the findings as proof of causality, which is not completely true. By this, I’m not saying that the contributions of this manuscript are irrelevant – I’m just trying to make sure no disinformation accidentally spreads from the study. Also, if I read the manuscript correctly, the study measured one specific behavioral retaliation response to phubbing (taking your own phone). Maybe some participants stop talking, leave the room, have an affair etc. The options are countless, so it might be worth mentioning them somewhere (at least among limitations).
  • There’s something unusual with the style of references

Generally, I like the manuscript and, with minor corrections, I believe it has its place in the journal. Well done!

Author Response

I would like to thank the authors for the opportunity to read their manuscript on narcissism and phubbing. My comments are below:

  • I like the intro (not all comments have to be negative ?)

Thank you for your positive comment.

  • If the RQ3 is exploratory, it might help readers to state that below the question.

We have added a statement about the exploratory nature of RQ3.

  • “Fewer students selected the dating option” <- students or participants?

This has now been corrected.

  • There’s a typo in “Pphubbing scale” in the methods section

The double p refers to partner phubbing. We have made this clearer now.

  • When reporting measures, the authors should also reflect on their metric properties (at least internal consistency if not model fit). This is mentioned later in the Results, but the methods section is where you have to justify the choice of measures, so it might fit better there.

In the Measures section, our aim is to report rather the use of the measures. Since information related to scale properties is already included in Table 1, we would prefer to present the reliability statistics there as well, for consistency and ease of reference.

  • What was the correlation between narcissistic rivalry and admiration?

We have added a correlation table to the manuscript that contains correlations between all variables. The correlation between admiration and rivalry is .35, which is consistent with Back et al. (2013).

  • An insignificant p has been bolded in Table 4

Thank you. We have now corrected this.

  • Causal language is a bit stronger in the Discussion than I’d prefer, considering potential Granger causality (e.g., rows 459-460). Some readers may be unfamiliar with research methodology and consider the findings as proof of causality, which is not completely true. By this, I’m not saying that the contributions of this manuscript are irrelevant – I’m just trying to make sure no disinformation accidentally spreads from the study.

Thank you for pointing this out. The language used throughout the discussion has been softened accordingly. 

  • Also, if I read the manuscript correctly, the study measured one specific behavioral retaliation response to phubbing (taking your own phone). Maybe some participants stop talking, leave the room, have an affair etc. The options are countless, so it might be worth mentioning them somewhere (at least among limitations).

We agree that individuals may respond to partner phubbing in a wide variety of ways beyond what we assessed, including more extreme or less visible forms of retaliation. In the current study, however, we examined a range of common responses drawn from previous literature on interpersonal conflict and phubbing—including curiosity, conflict, resentment, ignoring the behaviour, and retaliation. We agree that exploring a broader set of responses—including more covert or long-term behaviours—would be a valuable direction for future research, and we have now noted this in the Strengths, Limitations, and Future Directions section of the manuscript.

  • There’s something unusual with the style of references

The references were correctly formatted when originally submitted but have gone awry somewhere in the process. I will seek advice from the copy-editor as to how to correct them.

Generally, I like the manuscript and, with minor corrections, I believe it has its place in the journal. Well done!

Thank you very much!

Reviewer 2 Report

Comments and Suggestions for Authors

This manuscript presents a timely and relevant investigation into how narcissistic traits—specifically narcissistic admiration and narcissistic rivalry—influence individuals’ responses to perceived partner phubbing in romantic relationships. By employing a daily diary methodology, the authors offer a granular perspective on how trait narcissism interacts with modern digital behaviors, providing useful insight into both interpersonal dynamics and personality psychology in the digital age.

The use of daily self-reports is one of the study’s clear strengths, as it allows for a fine-grained analysis of behaviors and emotions as they unfold over time. The data are well-analyzed using multilevel modeling, and the interpretations are grounded in a solid review of the relevant literature. The distinction between admiration and rivalry within grandiose narcissism is especially valuable, and the authors make a meaningful contribution by unpacking these subtypes in the context of everyday conflict.

That said, there are several areas where the manuscript would benefit from further development—particularly in terms of theoretical framing, clarity, and scope.

First, while the study is methodologically sound, the theoretical context surrounding phubbing is relatively thin. The concept is treated primarily as a trigger, but its deeper relational implications—as a perceived rupture of mutual attention, or as a threat to intimacy and attachment—are not fully explored. Integrating theoretical perspectives from relationship science (e.g., attachment theory, relational maintenance models) would help position phubbing not just as a behavioral nuisance but as a phenomenon with emotional and symbolic weight. Similarly, the study could benefit from more reflection on the meaning of retaliation—whether it serves as boundary-setting, self-protection, or indirect communication.

Second, the discussion section would benefit from greater conceptual clarity and less repetition. Many of the findings are restated multiple times across sections, often without deepening the analysis. This gives the impression of circling rather than building toward insights. More focus on the “why” behind the findings—particularly around the mechanisms linking narcissistic traits to emotional responses—would strengthen the contribution. For example, the interpretation of curiosity as a mate-guarding strategy in narcissistic rivalry is compelling and could be extended to further connect personality and relational threat responses.

Third, while the sample size is commendable, the lack of diversity in gender and sexual orientation is a significant limitation. The paper acknowledges this, but it would benefit from a more substantial engagement with how gendered dynamics may influence both the perception of phubbing and the expression of narcissistic traits. A predominantly female, heterosexual sample limits the generalizability of the findings and should be more explicitly addressed not just as a methodological footnote but as a lens through which the findings must be interpreted.

Fourth, while the manuscript includes a brief mention of vulnerable narcissism in the limitations, its exclusion from the core analysis is a missed opportunity. Given that phubbing may be experienced as a form of rejection or abandonment, individuals high in vulnerable narcissism could be particularly affected. Even if not measured in this study, a speculative discussion of how this trait may function differently from grandiose narcissism would add depth and encourage future research.

Finally, the figures and tables, while informative, would benefit from more user-friendly design. The interaction graphs in particular are dense and not immediately interpretable to the average reader. Enhancing the visual clarity and adding succinct narrative captions would help the reader connect the statistical findings with their psychological meaning.

In sum, this is a promising and well-executed study that addresses an increasingly relevant topic in psychological and relational research. With more attention to theoretical framing, deeper discussion of the findings’ implications, and a stronger engagement with issues of diversity and generalizability, the manuscript will offer a more robust contribution to the literature on narcissism and digital-age relationship dynamics.

Author Response

This manuscript presents a timely and relevant investigation into how narcissistic traits—specifically narcissistic admiration and narcissistic rivalry—influence individuals’ responses to perceived partner phubbing in romantic relationships. By employing a daily diary methodology, the authors offer a granular perspective on how trait narcissism interacts with modern digital behaviors, providing useful insight into both interpersonal dynamics and personality psychology in the digital age.

The use of daily self-reports is one of the study’s clear strengths, as it allows for a fine-grained analysis of behaviors and emotions as they unfold over time. The data are well-analyzed using multilevel modeling, and the interpretations are grounded in a solid review of the relevant literature. The distinction between admiration and rivalry within grandiose narcissism is especially valuable, and the authors make a meaningful contribution by unpacking these subtypes in the context of everyday conflict.

That said, there are several areas where the manuscript would benefit from further development—particularly in terms of theoretical framing, clarity, and scope.

Thank you for highlighting the strengths of our paper.

First, while the study is methodologically sound, the theoretical context surrounding phubbing is relatively thin. The concept is treated primarily as a trigger, but its deeper relational implications—as a perceived rupture of mutual attention, or as a threat to intimacy and attachment—are not fully explored. Integrating theoretical perspectives from relationship science (e.g., attachment theory, relational maintenance models) would help position phubbing not just as a behavioral nuisance but as a phenomenon with emotional and symbolic weight.

We thank the reviewer for this insightful comment. We agree that phubbing can be conceptualised not only as a behavioural trigger but also as a symbolic disruption of relational norms, such as mutual attention, intimacy, and connection. In the introduction, we aimed to acknowledge this by referencing theories such as the displacement hypothesis and equity theory, which capture the emotional and relational significance of phubbing in terms of disrupted partner responsiveness and perceived imbalance in investment. We also discuss prior work that links phubbing with feelings of exclusion, jealousy, and diminished intimacy, thereby situating it within a broader relational framework. However, the primary focus of our paper was to explore individual differences—specifically narcissistic traits—in shaping emotional and behavioural responses to phubbing, rather than to provide an in-depth theoretical analysis of phubbing per se. To avoid diluting the paper’s central contribution, we deliberately tailored our theoretical framing to align with narcissistic motivations. We have however added content to make it clearer that phubbing is a symbolic act and not simply a nuisance and provide a deeper relational context that links to existing theories of relational functioning in the Emotional and behavioural responses to perceived partner phubbing section.

Similarly, the study could benefit from more reflection on the meaning of retaliation—whether it serves as boundary-setting, self-protection, or indirect communication.

We thank the reviewer for their suggestion to reflect further on the possible meanings of retaliation and whether it serves as boundary-setting, self-protection, or a form of indirect communication. Our manuscript already begins to explore these functions through the analysis of participants’ self-reported motivations for retaliatory phubbing and we outline the self-protective role of this for rivalrous narcissists in the discussion. However, we have now expanded this discussion to consider boundary setting and indirect communication in line with the reviewer’s helpful suggestion.

Second, the discussion section would benefit from greater conceptual clarity and less repetition. Many of the findings are restated multiple times across sections, often without deepening the analysis. This gives the impression of circling rather than building toward insights. More focus on the “why” behind the findings—particularly around the mechanisms linking narcissistic traits to emotional responses—would strengthen the contribution. For example, the interpretation of curiosity as a mate-guarding strategy in narcissistic rivalry is compelling and could be extended to further connect personality and relational threat responses.

We have attempted to reduce repetition and better highlight the why’s behind the findings. Throughout the discussion, we explicitly connect narcissistic rivalry with heightened sensitivity to ego threat and rejection, which we argue helps explain the retaliatory behaviours observed in our study. As for alternative responses to phubbing, we interpret curiosity about the partner’s phone use not simply as interpersonal interest, but as a potential mate-guarding response triggered by perceived relational threat—an interpretation grounded in prior research on narcissistic rivalry and defensive strategies (e.g., Back et al., 2013). We also differentiate between the strategic, esteem-regulatory behaviours of individuals high in narcissistic admiration (e.g., downplaying boredom as a motive) versus the more antagonistic, retaliatory motivations seen in rivalry. These patterns are not only highlighted but are interpreted through a lens of self-enhancement versus self-protection motives, in line with theoretical distinctions between the two narcissism subtypes.

Third, while the sample size is commendable, the lack of diversity in gender and sexual orientation is a significant limitation. The paper acknowledges this, but it would benefit from a more substantial engagement with how gendered dynamics may influence both the perception of phubbing and the expression of narcissistic traits. A predominantly female, heterosexual sample limits the generalizability of the findings and should be more explicitly addressed not just as a methodological footnote but as a lens through which the findings must be interpreted.

Thank you for this comment. We have revised the limitations section to engage more fully with how gender and relationship dynamics may shape responses to phubbing and the manifestation of narcissistic traits. We now explicitly note that gendered socialisation and norms may play a role in how phubbing is perceived or responded to, and that narcissistic behaviours may be expressed or judged differently across genders and relationship types. We also clarify that these contextual factors are important lenses through which to interpret our findings and should be explored further in future research with more diverse samples.

Fourth, while the manuscript includes a brief mention of vulnerable narcissism in the limitations, its exclusion from the core analysis is a missed opportunity. Given that phubbing may be experienced as a form of rejection or abandonment, individuals high in vulnerable narcissism could be particularly affected. Even if not measured in this study, a speculative discussion of how this trait may function differently from grandiose narcissism would add depth and encourage future research.

We have expanded on the discussion of how vulnerable narcissism may function differently from grandiose narcissism in the Strengths, Limitations, and Future Directions section.

Finally, the figures and tables, while informative, would benefit from more user-friendly design. The interaction graphs in particular are dense and not immediately interpretable to the average reader. Enhancing the visual clarity and adding succinct narrative captions would help the reader connect the statistical findings with their psychological meaning.

We have reported figures and tables in line with APA standards, however, we have added narrative captions which we hope makes the interaction graphs easier to interpret.   

In sum, this is a promising and well-executed study that addresses an increasingly relevant topic in psychological and relational research. With more attention to theoretical framing, deeper discussion of the findings’ implications, and a stronger engagement with issues of diversity and generalizability, the manuscript will offer a more robust contribution to the literature on narcissism and digital-age relationship dynamics.

Many thanks for your comments!

Reviewer 3 Report

Comments and Suggestions for Authors

Thank you for the opportunity to review this manuscript. I provide recommendations to help the paper reach its fullest potential.

  1. The literature review is filled with great information, but the flow and organization is hard to follow. I suggest adding some transition sentences and subheadings to address this concern.
  2. How many participants were recruited from Prolific and how many were recruited from social media posts? Were there any significant differences between both samples? Also, why were both of these approaches used? Were they conducted simultaneously, or was one used after another approach was not successful in getting enough participants?
  3. How were the data aligned across the daily surveys? At what time of the day did participants complete the surveys? How were the surveys provided to participants? Essentially, more details regarding the study methodology are needed.
  4. For longitudinal studies, Singer and Willett (2003) recommend at least three time points; however, for the current study, only two time points were required. Why was this decision made (i.e., can this approach be justified?). Would the results change if the minimum number of days of participation changed to 3?
  5. It may help to provide the correlations of study variables, or at least common on the correlations of study variables; potentially this information could be added to Table 1.

Author Response

  1. The literature review is filled with great information, but the flow and organization is hard to follow. I suggest adding some transition sentences and subheadings to address this concern.

We are glad you found the content interesting. To improve readability, we have added subheadings which we hope you find improves the flow.   

  1. How many participants were recruited from Prolific and how many were recruited from social media posts? Were there any significant differences between both samples? Also, why were both of these approaches used? Were they conducted simultaneously, or was one used after another approach was not successful in getting enough participants?

We have added information to the Participants section about the number of participants recruited via different sources and have added a footnote to highlight where there are any demographic or personality differences between samples. We initially tried to recruit using social media posts only (incentivised by a prize draw) but when data collection was extremely slow (perhaps not surprising given the longitudinal commitment), we used funds from research budgets to collect the remaining data using Prolific.

  1. How were the data aligned across the daily surveys? At what time of the day did participants complete the surveys? How were the surveys provided to participants? Essentially, more details regarding the study methodology are needed.

Additional details have been added to the procedure.

  1. For longitudinal studies, Singer and Willett (2003) recommend at least three time points; however, for the current study, only two time points were required. Why was this decision made (i.e., can this approach be justified?). Would the results change if the minimum number of days of participation changed to 3?

Although Singer and Willett (2003) recommend at least three time points for modelling individual growth trajectories in longitudinal research, our study’s analytic approach did not require estimating change over time but rather focused on between- and within-person associations across repeated observations. Specifically, with two time points, it is possible to examine change and associations between variables at each measurement occasion using multilevel modelling, which does not strictly require three or more waves when growth trajectories are not being estimated. Therefore, the decision to use two time points was guided by the study aims rather than growth modelling requirements. Increasing the minimum number of participation days to three would not fundamentally alter the analytic approach or the substantive conclusions, though it could provide more data per participant for estimating within-person effects.

  1. It may help to provide the correlations of study variables, or at least common on the correlations of study variables; potentially this information could be added to Table 1.

We have now included a separate Table (Table 2) with correlations of all study variables.

Many thanks for your comments!